# TRASH TO TREASURE: PAVING A NEW WAY FOR IMPROVING VIDEO UNDERSTANDING VIA COUNTERFACTUAL VIDEO GENERATION

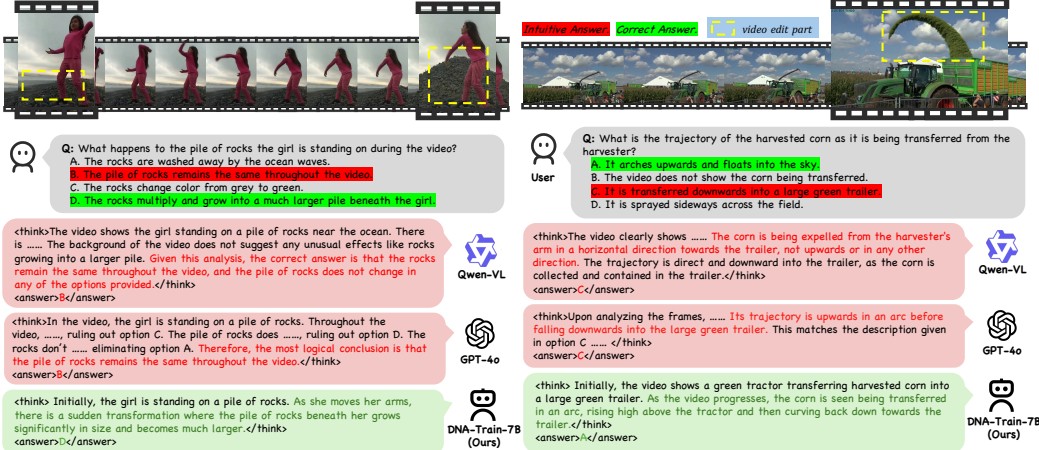

Figure 1: **Comparison of MLLMs on counterfactual video QA.** Each example shows key frames, the multiple-choice question, and the models' reasoning and answers. General models (Qwen-VL, GPT-4o) rely on language priors, picking plausible but visually incorrect answer. Our DNA-Train-7B model grounds reasoning in the video and selects the counterintuitive yet correct answer.

## ABSTRACT

Multimodal Large Language Models (MLLMs) have made remarkable progress in video understanding. However, they suffer from a critical vulnerability: an over-reliance on language priors, which can lead to "*visual ungrounded hallucination*", especially when processing counterfactual videos that defy common sense. This limitation, stemming from the intrinsic data imbalance between text and video, is challenging to address due to the substantial cost of generating and annotating counterfactual data. To address this, we introduce **DualityForge**, a novel counterfactual data synthesis framework that employs controllable, diffusion-based video editing to transform real-world videos into counterfactual scenarios. By embedding structured contextual information into the video editing and QA generation processes, the framework automatically produces high-quality QA pairs together with original–edited video pairs for contrastive training. Based on this, we build **DualityVidQA**, a large-scale video dataset designed to reduce MLLM hallucinations. Besides, to fully exploit the contrastive nature of our paired data, we propose **D**uality-**N**ormalized **A**dvantage **Train**ing (**DNA-Train**), a two-stage SFT-RL training regime where the RL phase incorporates $\ell_1$ normalization of advantages for each real-counterfactual pair, thereby enabling a more stable and efficient policy optimization. Experiments on **DualityVidQA-Test** demonstrate that our method substantially reduces model hallucinations on counterfactual videos, yielding a relative improvement of **24.0%** over the Qwen2.5-VL-7B baseline. Moreover, our approach achieves significant gains across both hallucination and general-purpose benchmarks, indicating strong generalization capability. We will open-source our dataset and code.

# 1 INTRODUCTION

Despite the remarkable advances in Multimodal Large Language Models (MLLMs) (Bai et al., 2025; Zhang et al., 2024c; Zhu et al., 2025a; Team Gemini et al., 2023; Achiam et al., 2023), studies have revealed a critical vulnerability of them: an over-reliance on language priors at the expense of genuine visual reasoning. This bias fosters "*visual ungrounded hallucination*", in which models generate responses driven more by learned common sense rather than the actual visual evidence (Li et al., 2025; Chen et al., 2024b). This issue becomes particularly severe when MLLMs process videos depicting counterfactual phenomena, as shown in Figure 1. When confronted with contents that defy such priors—such as an object vanishing or violating physical laws–MLLMs models often disregard the critical visual anomalies. As a result, they produce narratives that are linguistically plausible yet inconsistent with the actual events depicted in the video.

Most prior efforts to mitigate hallucinations in MLLMs have focused on modifying textual data (Chen et al., 2025b; Liu et al., 2024a; Yu et al., 2024), for example, altering video captions, to rebalance the distribution within the text modality. However, a primary cause of these hallucinations lies in the inherent data imbalance of MLLMs, where the scale and diversity of text far surpass those of video (Pi et al., 2024; Yao et al., 2025b). To address this, we advocate enhancing the model's visual perception through counterfactual data. However, this approach faces two key bottlenecks: (1) producing scalable counterfactual videos (*e.g.*, with visual effects) is both resource- and cost-intensive; and (2) generating high-quality QA pairs is hampered by a paradox: the models' own limited comprehension precludes reliable automatic data collection and annotation, resulting in a circular dependency that obstructs scalability.

Inspired by the recent advances in AI-Generated Content (AIGC) (OpenAI, 2023; 2024; Agostinelli et al., 2023), we introduce a novel data synthesis framework **DualityForge** that leverages controllable video editing (Liu et al., 2025; Mao et al., 2025), powered by diffusion models (Ho et al., 2020; Song et al., 2021), to transform real-world videos into counterfactual scenarios, such as erasing an object mid-clip to simulate a sudden disappearance. This type of method enables precise control over the generated events and, critically, embeds structured context (*e.g.*, event type, temporal location) into the editing process. This embedded context provides MLLMs with explicit cues to comprehend counterfactual phenomena, facilitating the automated, scalable creation of high-quality QA pairs. Furthermore, this process naturally yields paired data (original vs. edited videos), enabling an innovative contrastive QA training strategy. By requiring the model to provide different answers to identical questions for each video in a pair, we compel it to ground its reasoning in critical visual evidence instead of relying on language priors. Building upon this framework, we construct **DualityVidQA**, a large-scale video understanding dataset specifically designed to mitigate hallucinations in MLLMs, comprising 104K samples for SFT, 40K for RL, for a total of 144K training samples.

In terms of training methodology, we propose **D**uality-**N**ormalized **A**dvantage **Train**ing (**DNA-Train**), a two-stage regime—Supervised Fine-Tuning (SFT) followed by Reinforcement Learning (RL)—to mitigate hallucinations while preserving real-world performance. In the initial SFT stage, a hybrid dataset of real and counterfactual videos is used to enable the model to detect anomalies without compromising its performance on real videos. The subsequent RL stage further strengthens this capability. To balance the learning magnitude across different samples and avoid bias toward real videos, we apply $\ell_1$ normalization to the advantages for each real–counterfactual pair during RL, ensuring stable and balanced gradient updates, thereby better aligning with the contrastive nature of the training set and improving hallucination mitigation.

To evaluate model hallucinations and counterfactual video understanding capabilities, we introduce **DualityVidQA-Test**, a challenging benchmark of 600 manually-curated paired samples structured into 4 fine-grained counterfactual classes. Extensive experiments show our model achieves significant performance improvements not only on hallucination (*e.g.*, EventHallusion (Zhang et al., 2024a)) but also across leading general-purpose video understanding benchmarks (*e.g.*, TempCompass (Liu et al., 2024c), MVBench (Li et al., 2024)), demonstrating its robustness and broad applicability. In summary, the major contributions of our work are as follows:

- We propose **DualityForge**, the first counterfactual data synthesis framework that leverages diffusion-based controllable video editing with embedded structured priors to generate precise counterfactual scenarios, and, built upon it, we introduce **DualityVidQA**, a large-scale video un-

derstanding dataset (144K) for training and evaluating hallucinations in MLLMs, featuring paired videos with contrastive QA to systematically assess and mitigate model hallucinations.

- We introduce **DNA-Train**, a two-stage regime to compel the model to ground its reasoning in visual evidence. In addition, it $\ell_1$-normalizes the advantages for each real–counterfactual pair during RL, enabling a more stable and efficient policy optimization.

- Extensive experiments demonstrate that our approach achieves significant gains (24.0% on DualityVidQA-Test) across both hallucination and general benchmarks, indicating strong generalization capability and validating principle that generation can effectively enhance understanding.

## 2 RELATED WORKS

### 2.1 LANGUAGE PRIOR IN MLLMS

MLLMs inherit strong language priors from LLMs, which can lead to outputs that sound reasonable but conflict with visual evidence. Training-free contrastive decoding reduces this effect by contrasting the original logits with an auxiliary distribution (Li et al., 2022; Chuang et al., 2023), built via image masking, instruction perturbation, visual augmentation, or cross-modal conversion (Leng et al., 2024; Wang et al., 2024; Zhu et al., 2025b; Zhang et al., 2025a). However, this approach requires additional negative views, increases inference costs, is sensitive to hyperparameters, and does not allow updates to the base model. As a result, performance improvements on video and other temporal tasks are often unstable. Training-based methods construct specialized datasets (Liu et al., 2024a; Gunjal et al., 2024; Chen et al., 2025a), but this involves expensive prompting, filtering, annotation, and QA. In contrast, we propose an automated, scalable data synthesis framework that minimizes manual effort and applies naturally to video.

### 2.2 VIDEO UNDERSTANDING DATASETS

A large body of datasets support research on video understanding across tasks such as action recognition, temporal localization, retrieval, and question answering. Real-world collections include general action and activity corpora (*e.g.*, Kinetics (Kay et al., 2017), ActivityNet (Yu et al., 2019), EPIC-KITCHENS (Damen et al., 2018)), captioning and retrieval sets (*e.g.*, MSR-VTT (Xu et al., 2016), WebVid-10M (Bain et al., 2021), HowTo100M (Miech et al., 2019)). However, curating high-quality video-language annotations is expensive due to spatiotemporal complexity, which constrains the scale and granularity of labeled corpora. To mitigate these costs, recent studies leverage vision language models (VLM) to synthesize video language supervision at scale. LLaVA-Hound (Zhang et al., 2024b) and ShareGPT4Video (Chen et al., 2024a) prompt GPT-4 (Achiam et al., 2023) to generate instruction–response and question–answer (QA) pairs from videos, and LLaVA-Video (Zhang et al., 2024d) releases about 170K video–instruction examples via a scalable pipeline. These real video-based annotation pipelines show limitations in covering rare events, long-range dependencies and edited counter-commonsense scenarios, while facing category and domain imbalance issues.

### 2.3 VISUAL REINFORCEMENT LEARNING

Recent studies extend RL from text-only LLMs to multimodal settings to strengthen VLM understanding. Vision-R1 (Huang et al., 2025) addresses cold-start via a 200K multimodal CoT corpus and GRPO with strict formatting; R1-VL (Zhang et al., 2025b) introduces StepGRPO for step-wise rewards that better align intermediate steps with final answers; R1-ShareVL (Yao et al., 2025a) expands the question space and shares reasoning signals to mitigate sparse rewards. VL-Rethinker (Wang et al., 2025a) promotes slow thinking via selective replay and rethinking, and OpenVLThinker (Deng et al., 2025) interleaves SFT with RL to iteratively refine chains of thought. VLM-R1 (Shen et al., 2025) emphasizes training stability with rule-based objectives to curb reward hacking; ThinkLiteVL (Wang et al., 2025b) mines hard cases through Monte Carlo Tree Search; and VisionaryR1 (Xia et al., 2025) encourages grounding with a caption–reason–answer format and LLM-based caption rewards. Despite these advances, most methods still optimize textual traces (*e.g.*, CoT tokens) more than visual evidence, which limits robustness—especially against counterfactual or visually deceptive content. We stress that video understanding is not equivalent to textual reasoning: it requires discriminating visually plausible from counterfactual cues and aligning decisions with grounded evidence.

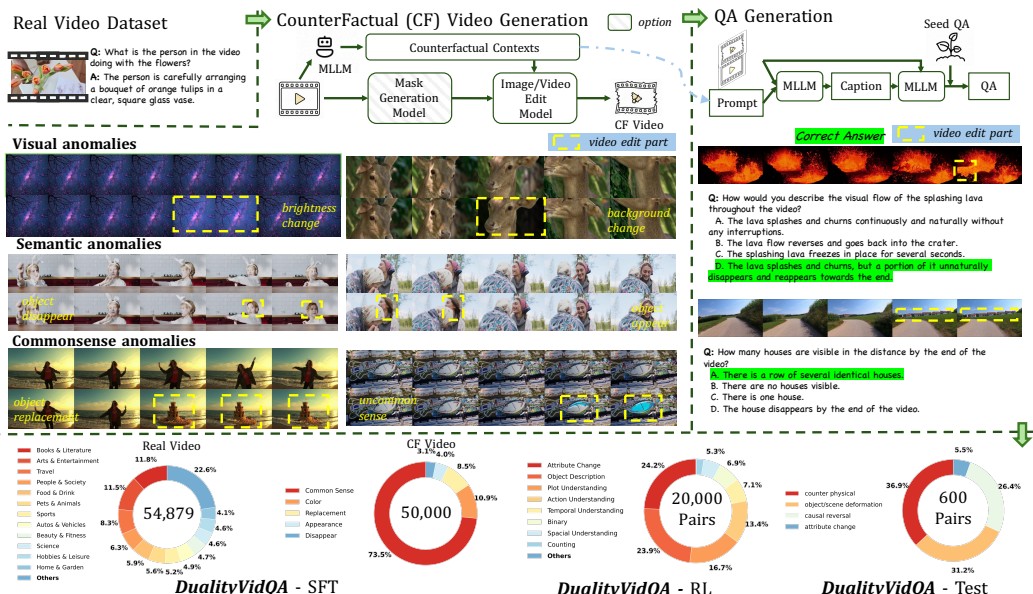

Figure 2: **Overview of the DualityForge framework and DualityVidQA dataset.** We begin with a web-sourced real-video dataset and apply a framework integrating MLLMs, grounding and segmentation modules, and image/video editing models to synthesize counterfactual (CF) videos with targeted visual, semantic, and commonsense alterations. Each real-CF video pair is paired with MLLM-generated questions using carefully designed prompts. The dataset comprises three splits: DualityVidQA-SFT with real and counterfactual video-QA pairs (54K + 50K) for SFT; DualityVidQA-RL with 20K shared-question contrastive video-answer pairs (one question, two real/CF instances) for RL; and DualityVidQA-Test (600 pairs), which shares the same contrastive structure as DualityVidQA-RL and covers diverse counterfactual categories.

## 3 DUALITYVIDQA

### 3.1 PROBLEM FORMULATION

Our work is motivated by a critical vulnerability in MLLMs: an inclination to favor dominant language priors over visual evidence (Leng et al., 2024; Huang et al., 2024). This bias, stemming from the data imbalance between massive text pre-training and comparatively limited video fine-tuning, leads to *visual ungrounded hallucination*. To mitigate this, our goal is to craft a large-scale video QA dataset comprising videos that depict visually salient counterfactual events. Each video is paired with questions designed to explicitly probe these anomalies, thereby encouraging the model to anchor its reasoning in visual evidence rather than linguistic bias. Formally, let $V$ be a video and $\mathcal{C}$ denote the context embodied in $V$. Our goal is to identify a *counterfactual* context $\mathcal{C}$ within a video $V$. Such a context is defined as one that creates a discrepancy between an answer derived from common-sense language priors and one derived from the actual visual evidence. We construct a question-answer pair $(Q, A)$ where the question $Q$ specifically probes this context $\mathcal{C}$, and $A = \{a_i\}_{i=1}^N$ represents the set of possible answers. To model this discrepancy, we distinguish between two conditional probabilities $P_*(a \mid \cdot)$ for any agent $* \in \{\text{human}, \text{LLM}, \text{MLLM}\}$: $P_*(a \mid Q)$ conditioned on the question, and $P_*(a \mid Q, V)$ conditioned on the question and video. Our objective is to find the most challenging contexts $\mathcal{C}$ that reveal an MLLM's hallucinations. A data sample is considered effective if it adheres to the following criteria, formalized as an optimization problem:

$$\max_{\mathcal{C}} \ D\left(P_{\text{MLLM}}\left(a \mid Q, V\right), P_{\text{human}}\left(a \mid Q, V\right)\right)$$
$$s.t. \ D\left(P_{\text{LLM}}\left(a \mid Q\right), P_{\text{human}}\left(a \mid Q\right)\right) \leq \epsilon \qquad (1)$$
$$D\left(P_{\text{human}}\left(a \mid Q\right), P_{\text{human}}\left(a \mid Q, V\right)\right) \geq \delta,$$

where $D$ is a divergence measure, $\epsilon$ and $\delta$ are small and large thresholds, respectively.

However, solving this optimization problem to construct the dataset at scale automatically remains intractable due to two primary bottlenecks:

1. **Data Scarcity**. Videos featuring naturally occurring counterfactual contexts $\mathcal{C}$ are inherently scarce and challenging to collect at scale.
2. **The Automation Paradox**. The MLLMs' perceptual blindness to these very phenomena prevents us from leveraging them to automate the data collection and annotation, resulting in a circular dependency that obstructs scalability.

To overcome these bottlenecks, we propose a paradigm shift that reframes the optimization from a search problem to a synthesis problem. Our approach leverages pre-defined counterfactual context $\mathcal{C}$ with a novel **duality**: first, it guides controllable diffusion-based video editing to transform a real-world video into a counterfactual video; second, it serves as a semantic blueprint to ground an MLLM's comprehension of the anomaly, unlocking a fully automated and scalable pipeline for high-quality QA generation, yielding QA pairs that adhere to the following principles:

$$\begin{cases} D\left(P_{\text{MLLM}}\left(a \mid Q, V\right), P_{\text{human}}\left(a \mid Q, V\right)\right) \geq \delta \\ D\left(P_{\text{MLLM}}\left(a \mid Q, V, \mathcal{C}\right), P_{\text{human}}\left(a \mid Q, V\right)\right) \leq \epsilon. \end{cases} \tag{2}$$

## 3.2 DUALITYFORCE

We categorize counterfactual context $\mathcal{C}$ into three hierarchical levels of increasing complexity. At the most fundamental level, **visual anomalies** refer to pixel-wise distortions (e.g., abnormal contrast, saturation) that degrade visual quality without changing scene semantics. Next, **semantic anomalies** disrupt object-level logic, introducing temporal inconsistencies such as object disappearance or substitution. Finally, **commonsense anomalies**, the most abstract category, encompass violations of real-world physics and plausibility, including unnatural deformations, impossible movements, or illogical agent interactions.

Based on the pre-defined $\mathcal{C}$ by MLLM, we propose a novel counterfactual data synthesis framework **DualityForce** (as shown in Figure 2) that transforms them into a comprehensive counterfactual dataset via a two-stage framework. The first stage involves employing a video editing model to embed the context $\mathcal{C}$ into a real-world source video, thereby generating the counterfactual video $V$. The second stage uses the same context $\mathcal{C}$, which acts as a semantic blueprint, enabling an MLLM to first generate an "oracle" caption and then self-produce a diverse set of grounded QA pairs (both multiple-choice and open-ended). Furthermore, we leverage the dual nature of our data (original vs. edited videos) to construct *shared-question* contrastive QA pairs. In this setup, the same question $Q$ is designed to yield different correct answers when applied to the original video ($V_{ori}$) versus the edited video ($V_{edit}$). This forces the VLM to ground reasoning in actual visual content and detect subtle changes, rather than relying on prior plausibility. Formally, this is achieved when:

$$D\left(P_{\text{MLLM}}\left(a \mid Q, V_{ori}\right), P_{\text{MLLM}}\left(a \mid Q, V_{edit}\right)\right) \geq \delta \tag{3}$$

To ensure the quality of our dataset, we implement a rigorous, model-based quality assurance process. This process validates the success of the video editing in the first stage and verifies the correctness of the generated QA pairs in the second stage. Built upon it, a large-scale, high-quality video understanding dataset, **DualityVidQA**, is constructed and partitioned into three dedicated splits: DualityVidQA-SFT (104K QA pairs from 25K original/edited video pairs), DualityVidQA-RL (20K *shared-question* contrastive video pairs; 40K QA pairs in total), totaling about 144K training QA pairs, and a human-annotated test set, DualityVidQA-Test (600 pairs). DualityVidQA-Test is further organized into four primary counter-commonsense scenarios derived from cluster analysis: **counter physical**, **object/scene deformation**, **attribute change**, and **causal reversal**. Further implementation details are available in Appendix A.

## 4 DNA-TRAIN

Motivated by the dual nature of our dataset, we present **DNA-Train**, a two-stage regime, SFT+RL, for mitigating hallucinations without sacrificing real-world performance, which employs a novel dual advantage normalization strategy to balance gradient updates. The structure of the **DNA-Train** is presented in Figure 3.

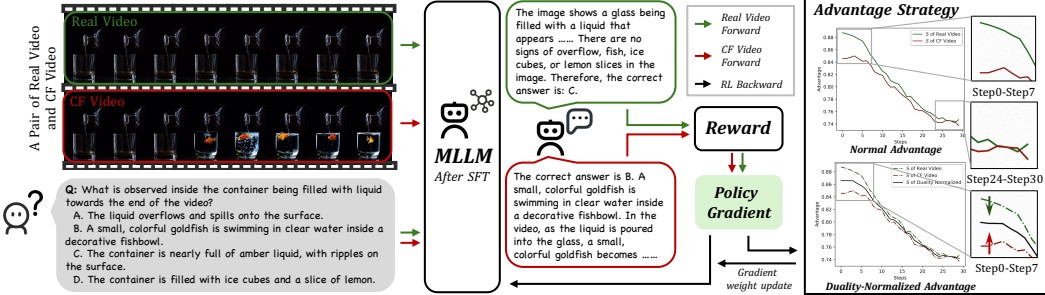

Figure 3: **Overview of DNA-Train framework.** We first perform SFT on our dual dataset to initialize the model. During RL, we sample a group of responses for both real and CF videos, compute their rewards based on task correctness, and calculate the $\ell_1$ norm of intra-group advantages. Finally, we normalize the advantages across the dual groups to ensure balanced gradients.

### 4.1 SUPERVISED FINE-TUNING

Our training begins with a supervised fine-tuning (SFT) stage on DualityVidQA-SFT. The primary objective is twofold: to instill the ability to recognize the embedded context $\mathcal{C}$ in edited videos ($V_{edit}$), while crucially maintaining robust performance on original, real-world videos ($V_{ori}$). To prevent the model from developing a bias towards either domain, we employ a balanced sampling strategy, ensuring each training batch contains an equal number of original and counterfactual samples. The training objective follows the cross-entropy loss: $\mathcal{L}_{\text{SFT}} = -\sum_{i=1}^{N} \log p_\theta(y_i|x_i)$, where $(x_i, y_i)$ represents the input-output pairs in our dataset, $\theta$ denotes the model parameters, and $p_\theta$ is the model's probability distribution over tokens.

### 4.2 REINFORCEMENT LEARNING

While SFT provides a foundational understanding, it lacks an explicit mechanism to directly penalize hallucinations and reward correct visual grounding. To further sharpen the model's reasoning, we introduce a second reinforcement learning (RL) stage. Unlike standard Reinforcement Learning from Human Feedback (RLHF) (Christiano et al., 2017; Ouyang et al., 2022), which relies on subjective, learned reward models, our task has a verifiable, ground-truth outcome, as the model must identify the sole correct answer from a list of choices. This singular ground truth makes our problem a natural fit for the Reinforcement Learning with Verifiable Rewards (RLVR) paradigm (Guo et al., 2025; Team Kimi et al., 2025), which uses a deterministic verifier $R : (\boldsymbol{q}, \boldsymbol{o}) \mapsto \mathbb{R}$ to provide unbiased, ground-truth rewards. Within the RLVR framework, algorithms like GRPO (Shao et al., 2024) have shown promise but often suffer from instability and entropy collapse on complex, long-chain-of-thought tasks—a common scenario in video QA. The more advanced DAPO algorithm (Yu et al., 2025) was specifically designed to overcome these limitations with enhancements for stable optimization over long trajectories. Therefore, we build the RL component of the advantage-normalization strategy upon the robust and scalable DAPO framework. Formally, for each QA pair $(\boldsymbol{q}, \boldsymbol{a})$, DAPO samples a group of outputs $\{\boldsymbol{o}_i\}_{i=1}^{G}$ with their corresponding rewards $\{R_i\}_{i=1}^{G}$, and then optimizes the policy via the following objective:

$$\mathcal{J}_{\text{DAPO}}(\theta) = \mathbb{E}_{(\boldsymbol{q},\boldsymbol{a})\sim\mathcal{D},\, \{\boldsymbol{o}_i\}_{i=1}^{G}\sim\pi_{\theta_{\text{old}}}(\cdot|\boldsymbol{q})}$$

$$\left[ \frac{1}{\sum_{i=1}^{G}|\boldsymbol{o}_i|} \sum_{i=1}^{G} \sum_{t=1}^{|\boldsymbol{o}_i|} \min\left( r_{i,t}(\theta)\hat{A}_{i,t}, \text{clip}\left(r_{i,t}(\theta), 1-\epsilon_{\text{low}}, 1+\epsilon_{\text{high}}\right) \hat{A}_{i,t} \right) \right], \quad (4)$$

$$\text{s.t.} \quad 0 < \left| \{\boldsymbol{o}_i \mid \text{is\_equivalent}(\boldsymbol{a}, \boldsymbol{o}_i)\} \right| < G$$

where

$$r_{i,t}(\theta) = \frac{\pi_\theta(\boldsymbol{o}_{i,t} \mid \boldsymbol{q}, \boldsymbol{o}_i, < t)}{\pi_{\theta_{\text{old}}}(\boldsymbol{o}_{i,t} \mid \boldsymbol{q}, \boldsymbol{o}_i, < t)}, \quad \hat{A}_{i,t} = \frac{R_i - \text{mean}(\{R_i\}_{i=1}^{G})}{\text{std}(\{R_i\}_{i=1}^{G})} \quad (5)$$

**Reward Design.** Our RL stage is guided by a dual-component reward signal derived from the *shared-question* contrastive QA pairs. The first component is a correctness reward, a binary score assigned for selecting the single right answer, which forces the model to capture subtle visual information. This is supplemented by a format reward, which encourages adherence to a prescribed reasoning structure. Together, these rewards optimize for both factual accuracy and the interpretability of the model's chain-of-thought process.

**Duality Advantages Normalization.** The gradient of $\mathcal{J}_{\text{DAPO}}(\theta)$ can be expressed[1] as:

$$\nabla_\theta \mathcal{J}_{\text{DAPO}}(\theta) = \mathbb{E}_{(q,a)\sim\mathcal{D},\ \{o_i\}_{i=1}^G\sim\pi_\theta(\cdot|q)}\left[\frac{1}{\sum_{i=1}^G |o_i|}\sum_{i=1}^G\sum_{t=1}^{|o_i|}\hat{A}_{i,t}\nabla_\theta\log\pi_\theta(o_{i,t}|q,o_{i,<t})\right]. \quad (6)$$

As shown in the equation, the DAPO gradient is modulated by the per-token advantage, $\hat{A}_{i,t}$. We use the $\ell_1$ norm of sequence-level advantages, $S = \sum_i \left|\hat{A}_i\right|$, as a proxy for the total learning signal magnitude from a group of responses, where $\hat{A}_i$ is the average of token-level advantages. With binary rewards, $S$ becomes a simple function of the group's average accuracy, $\overline{R}$:

$$S = |G|\sum_{i\in G}\left|\hat{A}_i\right| = 2\sqrt{(1-\overline{R})\overline{R}}, \quad (7)$$

This formulation reveals a critical property: the learning signal peaks for tasks of intermediate difficulty ($\overline{R} = 0.5$) and diminishes as tasks become trivial or impossible. As shown in Figure 3, we visualized $S_R$ and $S_{CF}$ under real ($G_R$) and counterfactual ($G_{CF}$) data. During the initial phase of training, the inherent accuracy gap between them creates a systematic imbalance in their learning signals, potentially destabilizing the training process. To counteract this, we introduce Duality-Normalized Advantage, which normalizes the advantages from each group to guarantee equal contribution to the gradient update. It computes scaling factors $\alpha_* = S_{target}/S_*$ (where $S_{target}$ is the mean of $S_R$ and $S_{CF}$) and applies them to their respective advantages. This elegant re-weighting scheme ($\hat{A}'_* = \alpha_*\hat{A}_*$) guarantees a balanced learning signal across disparate data types, fostering robust and equitable optimization. Further derivation details are available in Appendix B.

## 5 EXPERIMENT

### 5.1 EXPERIMENTAL SETUP

**Benchmarks.** We evaluate our model's performance across two categories of benchmarks: those focused on hallucination detection (DualityVQA-Test and EventHallusion (Zhang et al., 2024a)) and general video understanding benchmarks, including TempCompass (Liu et al., 2024c), MVBench (Li et al., 2024), TOMATO (Shangguan et al., 2024), and TVBench (Cores et al., 2024). Crucially, for DualityVQA-Test, we employ a stricter pairwise accuracy, where a sample is only counted if the model correctly answers for both the original and edited videos. Frame sampling adheres to each benchmark's standard protocol: 16 frames for DualityBench and TOMATO, 64 for TempCompass, and 8 for MVBench and TVBench.

**Implementation Details.** We leverage LLamaFactory (Zheng et al., 2024) for SFT and SWIFT (Zhao et al., 2025) for RL, applying both to the powerful Qwen2.5-VL base model. In the SFT stage, all models were trained for one epoch with a learning rate of $1 \times 10^{-6}$ and batch size of 4, using 8 H200 GPUs for 7B models and 16 for 32B/72B models. The RL stage maintained the same learning rate but with batch size of 64 and 16 sampled responses per prompt, running for 600, 60, and 20 steps for the 7B, 32B, and 72B models, respectively. For evaluation, we use greedy decoding (temperature=0) to ensure deterministic outputs.

### 5.2 EXPERIMENTAL RESULTS

Our analysis in Table 1 highlights a significant and consistent weakness across all evaluated MLLMs: a dramatic performance drop when moving from real to counterfactual videos. While leading closed-source models like GPT-4.1 and Gemini-2.5 Pro achieve 92% accuracy on "Real" videos, their performance on "Counterfactual" (CF) content is substantially lower. This gap is most evident in the

---

[1] We assume $\pi_{\theta_{\text{old}}} = \pi_\theta$ for simplicity.

Table 1: Performance comparison of different models on various anomaly categories (where CF indicates Counterfactual videos) from the DualityVidQA-test set. For each column, **bold** denotes the best score and underline denotes the second-best score.

| Model | Attribute Change | | | Causal Reversal | | | Counter Physical | | | Object/Scene Deformation | | | Overall | | |
|---|---|---|---|---|---|---|---|---|---|---|---|---|---|---|---|
| | Real | CF | Both | Real | CF | Both | Real | CF | Both | Real | CF | Both | Real | CF | Both |
| Random | 27.3 | 27.3 | 9.1 | 25.3 | 20.3 | 5.1 | 19.0 | 22.2 | 2.3 | 28.9 | 28.3 | 5.9 | 24.2 | 23.9 | 4.5 |
| *GPT Series* | | | | | | | | | | | | | | | |
| GPT-4o-mini(Hurst et al., 2024) | 84.8 | 51.5 | 36.4 | 89.9 | 58.2 | 50.0 | 91.4 | 53.4 | 48.9 | 95.2 | 62.6 | 59.9 | 91.8 | 57.4 | 51.9 |
| GPT-4o(Hurst et al., 2024) | 87.9 | 75.8 | 63.6 | 93.7 | 74.7 | 69.6 | 91.0 | 68.3 | 61.1 | 94.7 | 73.8 | 68.4 | 92.7 | 72.1 | 65.8 |
| GPT-4.1(OpenAI, 2025) | 84.8 | 84.8 | 69.7 | 89.2 | 81.6 | 73.4 | 86.4 | 68.8 | 59.7 | 87.2 | 76.5 | 65.2 | 87.3 | 75.5 | 65.6 |
| *Gemini Series* | | | | | | | | | | | | | | | |
| Gemini-2.5 Flash(Comanici et al., 2025) | 75.8 | 72.7 | 54.5 | 88.6 | 74.1 | 67.7 | 89.1 | 62.0 | 55.2 | 92.0 | 66.8 | 59.4 | 89.1 | 67.3 | 59.8 |
| Gemini-2.5 Pro(Comanici et al., 2025) | 84.8 | 81.8 | 69.7 | 91.8 | 88.0 | 80.4 | 92.8 | 78.3 | 73.3 | 94.1 | 75.9 | 71.1 | 92.5 | 80.3 | 74.3 |
| *Qwen Series* | | | | | | | | | | | | | | | |
| Qwen2.5-VL-7B(Bai et al., 2025) | 87.9 | 60.6 | 48.5 | 88.0 | 57.0 | 46.2 | 93.7 | 53.8 | 49.3 | 93.0 | 69.5 | 63.1 | 91.7 | 59.9 | 52.8 |
| Qwen2.5-VL-32B(Bai et al., 2025) | 87.9 | 54.5 | 45.5 | 94.3 | 68.4 | 63.3 | 95.5 | 43.0 | 39.4 | 96.8 | 59.4 | 56.1 | 95.2 | 55.4 | 51.3 |
| Qwen2.5-VL-72B(Bai et al., 2025) | 84.8 | 60.6 | 45.5 | 93.7 | 71.5 | 65.2 | 96.8 | 52.9 | 50.7 | 98.4 | 67.4 | 65.8 | 95.8 | 62.8 | 58.9 |
| *Other Models* | | | | | | | | | | | | | | | |
| VideoChat2-HD(Li et al., 2024) | 21.2 | 27.3 | 3.0 | 27.2 | 27.2 | 1.3 | 20.8 | 26.7 | 0.0 | 29.9 | 27.8 | 0.5 | 25.4 | 27.2 | 0.7 |
| LLaVA-Next-Video(Zhang et al., 2024c) | 57.6 | 33.3 | 9.1 | 67.1 | 29.7 | 13.9 | 69.2 | 31.7 | 16.3 | 71.1 | 42.8 | 21.4 | 68.6 | 34.7 | 16.9 |
| Video-LLaVA-7B(Lin et al., 2023) | 54.5 | 39.4 | 15.2 | 56.3 | 42.4 | 17.1 | 71.5 | 33.5 | 16.3 | 58.3 | 51.3 | 20.3 | 62.4 | 41.7 | 17.7 |
| *Ours* | | | | | | | | | | | | | | | |
| DNA-Train-7B | **97.0** | 72.7 | **72.7** | **94.3** | 74.1 | 69.0 | 94.6 | **83.3** | **79.2** | **98.4** | **82.9** | **81.3** | **95.8** | 80.1 | 76.8 |

Table 2: Performance comparison of different models on various benchmarks. For each task, **bold** denotes the best score and underline denotes the second-best score.

| Model | Hallucinations | | General Video Understanding | | | |
|---|---|---|---|---|---|---|
| | EventHallusion | DualityVidQA-Test | TempCompass | MVBench | TOMATO | TVBench |
| *Closed-source VLMs* | | | | | | |
| GPT-4o(Hurst et al., 2024) | **73.3** | 65.8 | **73.8** | 47.8 | **37.7** | 35.8 |
| *Open-source VLMs (∼ 7B size )* | | | | | | |
| VideoChat2-HD(Li et al., 2024) | 20.0 | 0.7 | 38.5 | 51.1 | - | 34.7 |
| LLaVA-Next-Video(Zhang et al., 2024c) | 12.1 | 16.9 | 44.7 | 42.2 | 20.1 | 38.2 |
| Video-LLaVA-7B(Lin et al., 2023) | 29.7 | 17.7 | 49.8 | 42.5 | 23.6 | 33.8 |
| Qwen2.5-VL-7B(Bai et al., 2025) | 33.5 | 52.8 | 71.4 | 62.6 | 26.8 | 51.7 |
| *Ours* | | | | | | |
| DNA-Train-7B | 61.3↑27.8 | **76.8**↑24.0 | 73.5↑2.1 | 63.8↑1.2 | 32.6↑5.8 | **53.0**↑1.3 |

overall results, where even the top-performing model, Gemini-2.5 Pro, drops from 92.5% (Real) to 80.3% (CF). This vulnerability is particularly acute in more challenging scenarios. For instance, in the "Counter Physical" category, most models struggle. However, our DNA-Train-7B demonstrates superior resilience, achieving a remarkable 79.2% in this category. As further confirmed in Table 2, our training methodology yields a dual benefit. First, DNA-Train-7B establishes itself as state-of-the-art in hallucination detection, achieving a top score of 76.8% on DualityVid-Test and massively outperforming other open-source models on EventHallusion. Critically, this specialization does not come at the cost of general video understanding. On the contrary, DNA-Train-7B consistently improves upon its base model (Qwen2.5-VL-7B) across all general benchmarks and remains highly competitive with, or even superior to, closed-source leaders like GPT-4o on benchmarks such as MVBench and TVBench. This ability to mitigate hallucinations while preserving broad video understanding capabilities marks a significant advance.

## 5.3 ABLATION STUDIES

**Ablations on Data Configurations**. As shown in Table 3, our ablation study on data configuration clearly demonstrates the necessity of our paired-data approach. Training on a single data type is markedly detrimental to our core task: using real data alone causes DualityVid-Test performance from the paired-data baseline of 52.8 to 29.0, while counterfactuals alone are even more damaging, with accuracy collapsing to 13.1. In contrast, the paired-data setting produces a clear synergistic effect—boosting DualityVid-Test performance to 70.6 and achieving the highest average improvement (+1.8) on the general video understanding benchmark. Intriguingly, training solely on counterfactual data improves general understanding (+1.7), suggesting that such data encourages the model to acquire more robust and generalizable visual representations.

Table 3: Ablation Study on Different Dataset Configurations.

| Setting | Hallucinations | | Avg Impr. | General Video Understanding | | | | Avg Impr. |
|---|---|---|---|---|---|---|---|---|
| | EventHallusion | DualityVidQA-Test | | TempCompass | MVBench | TOMATO | TVBench | |
| Base | 33.5 | 52.8 | - | 71.4 | 62.6 | 26.8 | 51.6 | - |
| Real Data | 29.4 | 29.0 | ↓ 7.9 | 72.4 | 61.5 | 23.5 | 50.9 | ↓ 2.1 |
| CF Data | 57.5 | 13.1 | ↓ 18.0 | 70.4 | 63.7 | 32.2 | 52.8 | ↑ 1.7 |
| Paired Data | 49.0 | 70.6 | ↑ 16.7 | 73.6 | 64.2 | 30.7 | 51.2 | ↑ 1.8 |

**Ablations on Duality-Normalized Advantages**. To isolate the effectiveness of our DNA strategy, we conducted an ablation study comparing it against strong RL baselines (GRPO, DAPO), starting from the same SFT-trained model. As shown in Table 4, DNA demonstrates clear superiority on the primary task of hallucination detection with an average improvement of 10.8. Furthermore, DNA also outperforms DAPO across every single general video understanding benchmark, demonstrating the effectiveness of our advantage normalization strategy.

Table 4: Ablation Study on Different RL Training Strategies.

| Method | Hallucinations | | Avg Impr. | General Video Understanding | | | | Avg Impr. |
|---|---|---|---|---|---|---|---|---|
| | EventHallusion | DualityVidQA-Test | | TempCompass | MVBench | TOMATO | TVBench | |
| Base | 57.8 | 58.7 | - | 72.2 | 63.7 | 31.6 | 51.5 | - |
| GRPO | 60.8 | 74.6 | ↑ 9.5 | 73.5 | 63.6 | 32.5 | 52.6 | ↑ 0.8 |
| DAPO | 60.6 | 74.8 | ↑ 9.5 | 73.0 | 63.0 | 32.5 | 52.6 | ↑ 0.5 |
| DNA | 61.3 | 76.8 | ↑ 10.8 | 73.5 | 63.8 | 32.6 | 53.0 | ↑ 1.0 |

**Ablations on Model Scales**. As shown in Table 5, our DNA-Train methodology consistently improves the Qwen2.5-VL model across all evaluated scales. The most substantial gains occur in hallucination detection, where the full DNA-Train boosts the average score by a substantial 25.9 points for the smallest model variant. Crucially, these gains are achieved without sacrificing general video understanding; in fact, our method delivers consistent gains on general benchmarks across all scales. In this process, SFT provides a strong foundation, while the subsequent RL step yields the largest boosts, particularly on the challenging DualityVid-Test benchmark. The smaller performance gain observed for the 72B model is primarily attributable to its reduced RL training schedule- 20 optimization steps compared to 60 for the 32B and 600 for the 7B -an intentional trade-off necessitated by computational resource constraints. Additional ablation studies are detailed in the Appendix C.

Table 5: Ablation Study on different model sizes of Qwen2.5-VL.

| Type | Model | Hallucinations | | Avg Impr. | General Video Understanding | | | | Avg Impr. |
|---|---|---|---|---|---|---|---|---|---|
| | | EventHallusion | DualityVidQA-Test | | TempCompass | MVBench | TOMATO | TVBench | |
| 7B | Base | 33.5 | 52.8 | - | 71.4 | 62.6 | 26.8 | 51.6 | - |
| | + SFT | 57.8 | 58.7 | ↑ 15.1 | 72.2 | 63.7 | 31.6 | 51.5 | ↑ 1.7 |
| | + SFT+RL | 61.3 | 76.8 | ↑ 25.9 | 73.5 | 63.8 | 32.6 | 53.0 | ↑ 2.6 |
| 32B | Base | 34.0 | 51.2 | - | 75.2 | 61.5 | 31.0 | 51.5 | - |
| | + SFT | 55.6 | 60.0 | ↑ 15.2 | 74.1 | 61.7 | 33.6 | 54.3 | ↑ 1.1 |
| | + SFT+RL | 58.8 | 60.8 | ↑ 17.2 | 74.2 | 61.9 | 34.6 | 54.7 | ↑ 1.4 |
| 72B | Base | 54.6 | 58.9 | - | 77.6 | 64.8 | 36.3 | 55.5 | - |
| | + SFT | 64.6 | 68.3 | ↑ 9.7 | 78.0 | 65.7 | 35.7 | 56.9 | ↑ 0.5 |
| | + SFT+RL | 65.4 | 69.4 | ↑ 10.7 | 78.3 | 65.9 | 36.5 | 57.3 | ↑ 0.9 |

# 6 CONCLUSION

In this work, we address the critical issue of visual hallucinations in MLLMs, which stems from an over-reliance on language priors when processing visual content. To this end, we introduce **DualityForge**, a novel framework that uses controllable video editing to generate a large-scale (144K) contrastive dataset, **DualityVidQA**, comprising paired real and counterfactual videos. Building on this, we propose **DNA-Train**, a two-stage regime that $\ell_1$-normalizes advantages per real counterfactual pair during RL to ensure balanced training and compel the model to ground its reasoning in visual evidence. Extensive experiments demonstrate that our approach not only significantly reduces hallucinations but also boosts performance on general video understanding benchmarks. By turning "trash" data that defies common sense into a "treasure" for robust training, we pave a new and promising way towards more reliable and visually-grounded video understanding.

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

# A    DATSET DETAIL

We categorize video anomalies into three levels: Visual anomalies refer to pixel-wise distortions, including abnormal contrast, saturation, brightness, blurring, and local distortions, etc., which primarily affect visual quality without explicit semantic alteration. Semantic anomalies involve violations of scene semantics, such as object disappearance, unexpected object emergence, and object substitution, which result in temporal inconsistencies. Commonsense anomalies capture more abstract and holistic violations involving spatio-temporal or physical implausibility, such as unnatural deformations, implausible object movements, unreasonable interaction and human motion anomalies, etc.

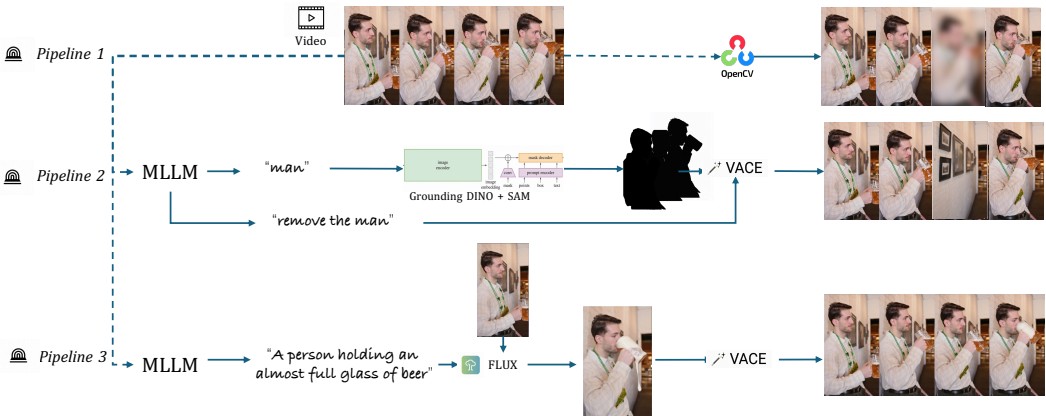

Figure A.1: Overview of Counterfactual video edit framework. There are 3 pipeline are shown: 1. Visual Anomaly: we use opencv to edit the video in pixel-level. 2. Semantic Anomaly: we use VACE to edit the video in object-level. 3. Common Sense Anomaly: we use MLLM to generate the edit instruction, then use FLUX-Kontext to edit the first frame to end frame, finally use VACE to interpolate the video.

## A.1    DUALITYFORGE

Table A.1: Definitions of video anomaly categories.

| Category | Definition |
|---|---|
| **Visual** | Pixel-wise distortions that primarily affect visual quality without explicit semantic alteration. These include abnormal contrast, saturation, brightness, blurring, and local distortions. |
| **Semantic** | Violations of scene semantics, such as object disappearance, unexpected object emergence, and object substitution, resulting in temporal inconsistencies. |
| **Commonsense** | Abstract and holistic violations involving spatio-temporal or physical implausibility (*e.g.*, unnatural deformations, implausible object movements, unreasonable interactions, and human motion anomalies). |

**Video Source.** To improve video-editing quality and dataset diversity, we adopt two widely used public datasets Pexels (Corran, 2022) and OpenVid (Nan et al., 2024) which are commonly employed in video-generation research. From OpenVid, we randomly sample 3,000 videos from each of the 20 most populated categories, yielding a candidate pool of 61,591 clips. From Pexels, we additionally sample 36,333 clips, for a total of 97,924 videos.

**Visual anomalies.** We employ OpenCV to synthesize visual anomalies within the video data. We divide visual anomalies into **entire-frame** level, **region** level, and **object** level. To introduce anomalies, we randomly select a temporally consistent segment in which to insert visual perturbations. At

the object level, we first extract all noun entities present in the video and randomly select one object. Then we utilize Grounding DINO(Liu et al., 2024b) and SAM(Ravi et al., 2024) to localize the position of the selected object, on which the visual anomaly synthesis operation is performed.

**Semantic anomalies.** We categorize semantic anomalies to include both the temporal instability of entities (*e.g.*, unexpected appearance, disappearance, or substitution) and appearance-level abnormalities (such as unreadable text or blurred faces). To enable controlled injection of anomalies into the video while keeping the other part unchanged, we utilize the advanced video editing model, VACE(Jiang et al., 2025), to edit the specific area in the video.

**Common sense anomalies.** We categorize anomalies that contradict common sense into the following types: violations of physical laws, causal inconsistencies, material abnormalities, and abnormal human movements. To introduce the first three types of anomalies into videos, we first employ a Multimodal Large Language Model (MLLM) to analyze the visual elements within an image and generate an editing instruction targeting the anomaly. Next, we use FLUX-Kontext(Batifol et al., 2025) to edit the image according to this instruction. After validating the edited image, we create a video by performing frame interpolation with VACE using the original and edited image pair.

Finally, we collect a total of 133, 168 videos with anomalies. The statistics of video types are shown in Table A.2. This takes around 40k GPU hours on NVIDIA H20 GPUs.

Table A.2: Video dataset type statistics

| Type | Count |
|------|-------|
| color | 27353 |
| replacement | 9961 |
| appearance | 6092 |
| disappear | 5016 |
| common sense | 86746 |
| All | 133168 |

## A.2 DUALITYVIDQA

**Training Data Construction.** To enhance VLM counter-commonsense reasoning while preserving general VideoQA performance, we adopt a two-stage training framework: Supervised Fine-Tuning (SFT) and Reinforcement Learning (RL). For each stage, we curate a tailored dataset to support its specific training objective. We conducted **two rounds** of data curation to ensure optimal training quality. In our first round, we constructed initial datasets for both SFT and RL stages. We generated 200k QA pairs from 80k videos. After analyzing the training performance, we observed that samples with zero reward were predominantly associated with failed video edits where no meaningful visual changes were created. Thus, we use the first stage trained model to filterout around 30% of the samples with zero reward and low-quality video. This insight led us to create a refined dataset through the following process:

Table A.3: Question type frequency statistics

| QA Type | Real Video | Counterfactual Video |
|---------|-----------|---------------------|
| Multiple Choice | 12210 | 10224 |
| Open-Ended | 42669 | 39776 |
| All | 54879 | 50000 |

**(1) SFT** data construction through two stages: dense captioning and question-answer (QA) generation. During dense captioning, a red box is used to indicate the anomaly region, and video editing metadata is provided to the model to generate detailed, high-coverage captions under controlled conditions. The detailed prompt is **Dense Caption Prompt Template**.During QA generation, we followed LLaVA-Video, categorizing questions into 16 types and using GPT-5 and Gemini 2.5 Pro to generate questions and answers based on video content and dense captions. To ensure diversity

Table A.4: 16 Question type frequency statistics with descriptions

| QA Type | Real Video | Counterfactual Video | Description |
|---|---|---|---|
| Attribute Change | 1436 | 8674 | Questions about changes in attributes of objects or characters between scenes or frames. |
| Binary | 2009 | 1009 | Involves yes or no questions related to the video content. |
| Camera Direction | 1601 | 4887 | Tests understanding of the camera's movement or shooting direction within the video. |
| Causal | 737 | 216 | Focuses on explaining actions/events, determining intentions of actions or causes for events. |
| Count | 363 | 438 | Tests ability to count instances of objects, people, or actions. |
| Description Human | 15360 | 4324 | Involves describing actions or attributes of people. |
| Description Object | 8450 | 4404 | Assesses ability to describe attributes of objects. |
| Description Scene | 19067 | 8317 | Assesses ability to describe the major scene of the video. |
| Fine-grain Action Understanding | 811 | 1303 | Creates questions challenging comprehension of subtle actions. |
| Non-Existent Actions with Existent Scene Depictions | 29 | 113 | Tests ability to identify actions that did not occur despite related scene elements being present. |
| Object Direction | 420 | 3374 | Tests understanding of the movement or facing direction of objects within the video. |
| Plot Understanding | 981 | 151 | Challenges ability to interpret the plot in the video. |
| Spatial | 2074 | 8641 | Tests ability to perceive spatial relationships between observed instances in a video scene. |
| Speed | 221 | 998 | Involves estimating or comparing the speed of moving objects or actions. |
| Temporal | 768 | 2789 | Designed to assess reasoning about temporal relationships between actions/events. |
| Time Order Understanding | 552 | 362 | Tests comprehension of the chronological order of events or actions in the video. |
| All | 54879 | 50000 | Aggregate counts for all question types. |

and stability, we sampled 5,000 examples from LLaVA-Video's 170k dataset as a pool, randomly selecting three same-category examples at each generation step as in-context references to maintain stylistic consistency and content diversity. Finally, we curated 25K real videos and 25K edited videos, generating 100K QA pairs with an 8:2 ratio of open-ended to multiple-choice items. Then we use GPT-4o to classify each QA into question types based on the LLaVA-Video taxonomy. The qa detail statistics are shown in Table A.4 and Table A.3. The examples of SFT QA are shown in Figure A.2.

**(2) RL** data construction centers on creating *shared-question* counterfactual QA pairs: for each *real* and *edited* video pair, we design the same question and identical answer candidates, but the correct answer differs between the two videos. This forces the VLM to ground reasoning in actual visual content and detect subtle changes, rather than relying on prior plausibility. We construct the RL dataset using Gemini2.5-Pro, which generates counterfactual QA pairs from video captions by identifying visual differences. The prompting strategy follows the **Counterfactual Video QA Prompt Template**. In total, we curate 20K counterfactual QA pairs as the RL training dataset. The examples of RL QA are shown in Figure A.3.

Table A.5: Counterfactual video category statistics in DualityVidQA-Test

| Tag | Count |
|---|---|
| causal reversal | 158 |
| counter physical | 221 |
| object/scene deformation | 187 |
| attribute change | 33 |
| All | 599 |

**(3) Test Set.** We construct a high-quality test set, DualityVidQA-Test, to evaluate counter-commonsense reasoning. Firstly, we sample around 2000 pairs from our paired video pool. Then, we employ Gemini 2.5 Pro to generate candidate based on video content and dense captions. The prompt is **RL Question Generation Prompt**. The we employ 3 human annotators and 3 expert reviewers to filter and refine the generated QA pairs, ensuring each question is valid, unambiguous, and answerable based on the video content.

The final test set consists of 600 real-counterfactual video pairs, each with a shared question and options but different answers. We then cluster the test set into 12 categories, then manually cluster them into 4 major categories: counter physical, object/scene deformation, causal reversal, and

attribute change. The statistics of counterfactual video categories are shown in Table A.5. The examples of test QA are shown in Figure A.4.

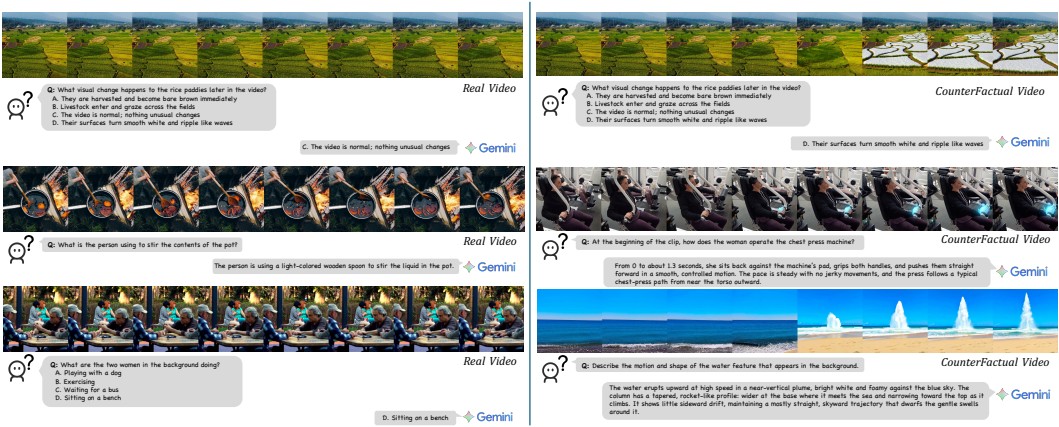

Figure A.2: Examples of DualityVidQA-SFT. We show the real video and counterfactual video pair and the question and answer pair generated based on the counterfactual video.

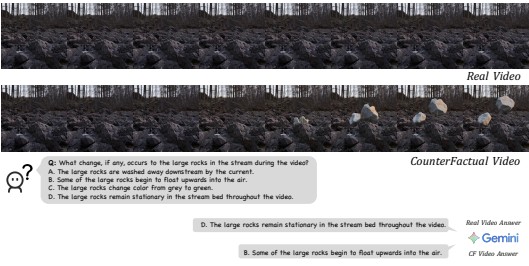

Figure A.3: Examples of DualityVidQA-RL. We show the real video and counterfactual video pair and the generated question and answer.

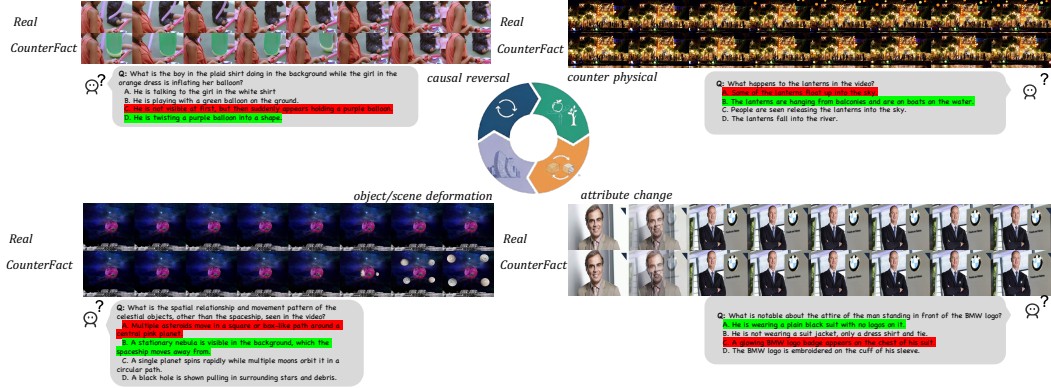

Figure A.4: Examples of DualityVidQA-Test. We show the real video and counterfactual video pair and the generated question. Answers for the counterfactual video are shown in red, and answers for the real video are shown in green.

## A.3 PROMPT TEMPLATE

### A.3.1 DENSE CAPTION PROMPT TEMPLATE

---

**Dense Caption Prompt Template**

You are a professional video understanding and visual anomaly detection expert. Please generate a description for the given video.

Given **[num_frames]** uniformly sampled frames from the video (total duration approximately **[video_duration]** seconds), the time periods are **[time_list]**, please generate a detailed description in chronological order.

I will use a red box to figure out the anomaly region.

- anomaly_type: **[anomaly_type]**
- region_type: **[region_type]**
- region_name: **[region_name]**
- anomaly_start_time(s): **[anomaly_start_time]**
- anomaly_end_time(s): **[anomaly_end_time]**

Please pay special attention to the following points:

1. Describe in detail the important objects, actions, and relationships in each time period of the video, following chronological order, and merge the same content to ensure the dense caption is efficient and clear.

2. Carefully analyze and point out any visual anomalies, such as:
    - Unnatural changes in object appearance (distortion, warping, blurring, etc.)
    - Perspective distortion or geometric deformation in specific regions
    - Discontinuities or unnatural transitions in object edges
    - Abnormal changes in texture or color
    - Unnatural changes in lighting effects
    - Anomalous behavior in specific regions of the video (e.g., lens, objects)

3. For each detected anomaly, please specify in detail:
    - The exact time period when the anomaly appears
    - The specific region or object affected by the anomaly
    - You need to convert the anomaly parameters into natural language descriptions (do not output values like 'saturation factor is xx', and do not output specific region coordinates).

4. If there are other anomalies, such as blurring, missing content, unrecognizable scenes, etc., please clearly point them out in the relevant paragraphs.
    But don't need mention red box in the description.

The output format should be JSON, including the following content:

```
{
    "spatial_location": "[region_name]",
    "merged_timestamps": ["[0.0s - ...]", "[... - ...]"],
    "dense_captions": [
        "[0.0s - ...]: ...",
        "[... - ...]: ...",
        "...",
        "[... - video_end_time]: ..."
    ]
}
```

---

## A.3.2 COUNTERFACTUAL VIDEO QA GENERATION PROMPT TEMPLATE

---

**Counterfactual Video Multiple Choice and Open-Ended Question Generation Prompt**

**Task** Given a detailed description that summarizes the content of the generate-video, generate question-answer pairs to build LLM training data.

**Reference Examples:** Here is one question dimension and its explanation and example question-answer pairs for reference:

**Question Type**: [question_type]

Example 1:

```
## caption-1: [Video description]
## question-1: [Question text]
## answer-1: [Answer text]
```

Example 2:

```
## caption-2: [Video description]
## question-2: [Question text]
## answer-2: [Answer text]
```

Example 3:

```
## caption-3: [Video description]
## question-3: [Question text]
## answer-3: [Answer text]
```

You need to generate similar question-answer pairs like the examples.

**Guidelines For Question Generation:**

- Please formulate questions using only objectively observable information, without presupposing or emphasizing any abnormal, strange, or logically impossible phenomena in the questions themselves.

- The questions should be neutral and natural, while the answers may accurately describe the observed phenomena.

- Each multiple-choice question should have 4 options (A, B, C, D), with only one correct answer.

- The answer must be correct with respect to the video visual content.

- For abnormal object/event questions, include an option stating "The video is normal" as a distractor.

- Generate 1-4 question-answer pairs.

- Do not mention people's reactions to abnormal phenomena.

- For open-ended questions, provide detailed descriptions including speed and direction of actions/camera movements.

**Input:** Dense Caption: [dense_caption]

**Output Format:** Your output should be formatted as a JSON file:

For Multiple Choice Questions:

```
[{
    "Question": "<question-1>",
    "Options": ["<option-0>", "<option-1>",
                "<option-2>", "<option-3>"],
    "Answer": "index of correct option"
}]
```

For Open-Ended Questions:

```
[{
    "Question": "<question-1>",
    "Answer": "<a detailed answer-1>"
}]
```

---

### A.3.3 REAL VIDEO PROMPT TEMPLATE

---

**Real Video Multiple Choice and Open-Ended Question Generation Prompt**

**Task:** Given a detailed description that summarizes the content of video, generate question-answer pairs to build LLM training data.
**Reference Examples:**
**Question Type**: [question_type]
For Multiple Choice:

```
## caption-1: [Video description]
## question-1: [Question text]
## options-1: [A. Option1, B. Option2, C. Option3, D. Option4]
## answer-1: [Correct answer]
```

For Open-Ended:

```
## caption-1: [Video description]
## question-1: [Question text]
## answer-1: [Detailed answer]
```

You need to generate similar question-answer pairs like the examples.
**Guidelines For Question Generation:**
**For Multiple Choice Questions:**

- Generate appropriate multiple-choice question-answer pairs based on the description

- Each question should have 4 options (A, B, C, D)

- Only one option should be correct

- Other options should be plausible distractors

- Distractor options must be reasonable, relevant to the question, and not obviously wrong

**For Open-Ended Questions:**

- Generate appropriate question-answer pairs based on the description

- Answers should be detailed and comprehensive

**General Guidelines:**

- Generate 1-4 question-answer pairs

- Questions should focus on observable content in the video

- Maintain natural and objective question formulation

**Output Format:**
For Multiple Choice Questions:

```
[{
    "Question": "<question-1>",
    "Options": ["<option-0>", "<option-1>",
                "<option-2>", "<option-3>"],
    "Answer": "index of correct option"
}]
```

For Open-Ended Questions:

```
[{
    "Question": "<question-1>",
    "Answer": "<a detailed answer-1>"
}]
```

---

### A.3.4 COUNTERFACTUAL VIDEO QA PROMPT TEMPLATE

---

**RL Question Generation Prompt**

**Task:** Given two captions — **TRUE CAPTION** (original video description) and **MOCK CAPTION** (edited video description after applying an **edit instruction**) — design a question that can be answered differently for the TRUE and MOCK videos. The goal is to produce high-quality, dimension-specific question-answer pairs for training multimodal models.

**Reference Example:**
TRUE CAPTION: The man places a cake on the table and lights the candles. MOCK CAPTION: The man places a cake on the table without lighting any candles. Edit Instruction: Remove the candle lighting action.
Question: What does the man do with the cake after placing it on the table? Answer for TRUE: He lights the candles on the cake. Answer for MOCK: He leaves the cake as it is without lighting candles. Wrong Answers: ["He cuts the cake into slices", "He puts the cake back into the oven"]

**Guidelines for Question Generation:**

**Core Requirements:**

- Base questions strictly on differences between the TRUE and MOCK videos.
- Do not refer to or mention captions directly in the question.
- No timestamps or meta-information in the question.
- Use the provided **edit instruction** as a design hint.
- Questions must belong to one of the predefined **task dimensions**.
- If no suitable question for the chosen dimension, output an empty question string.
- Wrong answers must be incorrect for both videos, but still plausible.
- Generate answers for each video independently without inferring from the other.

**Available Dimensions:** Refer to the predefined TASK_EXAMPLES set for dimensions and descriptions.

**Output Format:** The result must be **valid JSON** with the following structure:

```
{
    "dimension": "<task dimension>",
    "question": "<generated question>",
    "answers_for_true_caption": ["<answer based on TRUE CAPTION>"],
    "answers_for_mock_caption": ["<answer based on MOCK CAPTION>"],
    "wrong_answers": ["<wrong answer 1>", "<wrong answer 2>", ...]
}
```

---

## B DERIVATION

Here we show the derivation of Eq 7.

We consider the case where the reward values $R_i$ are binary, i.e.,

$$R_i \in \{0, 1\}. \tag{8}$$

Let $|G|$ be the size of the group, and let

$$\overline{R} = \frac{1}{|G|} \sum_{i \in G} R_i \tag{9}$$

denote the accuracy of the group (i.e., the fraction of correct responses).

**Standard Deviation of rewards.**

$$std(\{R_i\}_{i=1}^G) = \sqrt{\frac{\overline{R} \cdot |G| \cdot (1 - \overline{R})^2 + (1 - \overline{R}) \cdot |G| \cdot (0 - \overline{R})^2}{|G|}} \tag{10}$$

$$= \sqrt{\overline{R} \cdot (1 - \overline{R})}$$

Following Eq 5, the magnitude of the advantage is therefore:

$$|\hat{A}_i| = \begin{cases} \frac{1-\overline{R}}{\sqrt{\overline{R}\cdot(1-\overline{R})}}, & \text{if } r_i = 1, \\ \frac{\overline{R}}{\sqrt{\overline{R}\cdot(1-\overline{R})}}, & \text{if } r_i = 0. \end{cases} \qquad (11)$$

**Sum of $\ell_1$ norm.** The sum of $\ell_1$ norm of $\hat{A}_i$ over the group is:

$$S = \frac{1}{|G|}\sum_{i \in G}|\hat{A}_i| = \frac{1}{|G|}\left[|G|\cdot\overline{R}\cdot\frac{1-\overline{R}}{\sqrt{\overline{R}\cdot(1-\overline{R})}} + |G|\cdot(1-\overline{R})\cdot\frac{\overline{R}}{\sqrt{\overline{R}\cdot(1-\overline{R})}}\right]$$
$$= 2\sqrt{\overline{R}\cdot(1-\overline{R})} \qquad (12)$$

## C SUPPLEMENTAL EXPERIMENTS

Table C.1: Effect of model type.

| Model | Stage | Hallucinations | | Avg Impr. | General Video Understanding | | | | Avg Impr. |
|---|---|---|---|---|---|---|---|---|---|
| | | EventHallusion | DualityBench | | TempCompass | MVBench | TOMATO | TVBench | |
| Qwen2.5vl 7B | Base | 33.5 | 52.8 | - | 71.4 | 62.6 | 26.8 | 51.6 | - |
| | +SFT | 57.8 | 58.7 | ↑ 15.1 | 72.2 | 63.7 | 31.6 | 51.5 | ↑ 1.7 |
| | +SFT+RL | 61.3 | 76.8 | ↑ 25.9 | 73.5 | 63.8 | 32.6 | 53.0 | ↑ 2.6 |
| LLaVA-Next-Video | Base | 12.1 | 16.9 | - | 44.7 | 42.2 | 20.1 | 38.2 | - |
| | +SFT | 53.3 | 57.2 | ↑ 35.7 | 51.9 | 45.3 | 22.6 | 36.3 | ↑ 2.7 |
| | +SFT+RL | 51.9 | 67.6 | ↑ 42.0 | 52.9 | 46.8 | 21.4 | 38.7 | ↑ 3.7 |

**Influence of Different Model Architectures**. We conducted experiments on two open-source MLLMs, namely LLaVA-Next-Video(Zhang et al., 2024c) and Qwen2.5-VL as summarized in Table C.1. The results indicate that, after training on **DualityVidQA** with our **DNA-Train**, both models consistently outperform their baseline versions across the evaluated metrics. On the Qwen2.5-VL 7B model, the application of DNA Train results in average performance improvements of 25.9 and 2.6 compared with the baseline on hallucination and general benchmarks, respectively. Similarly, on LLaVA-Next-Video, which starts from a lower baseline, the performance gain is 42.0 and 3.7 on hallucination and general benchmarks. These results indicate that our DNA-Train method not only enhances counterfactual reasoning ability significantly, especially on DualityBench, but also improves general video understanding performance across different model architectures, demonstrating its robustness and broad applicability.

## D USE OF LLMS

We utilized large language models (LLMs) to assist in refining the phrasing of certain sentences in this manuscript. Their use was limited to improving clarity and readability; all ideas, analyses, and conclusions are our own.

