# OpenReview forum: "Trash to Treasure: Paving a New Way for Improving Video Understanding via Counterfactual Video Generation"
_ICLR.cc/2026/Conference — ICLR 2026 Conference Withdrawn Submission_

### Official Review · Reviewer_348k · 2025-10-21

**Soundness:** 3
**Presentation:** 3
**Contribution:** 3
**Rating:** 4
**Confidence:** 5

**Summary:**

The paper introduces DualityForge, a generative augmentation framework designed to reduce hallucinations in multimodal large language models (MLLMs). The core idea is to synthesize counterfactual videos using an off-the-shelf text-to-video diffusion model, which transforms real videos into controlled perturbations across three categories, visual (e.g., pixel-level distortions), semantic (e.g., appearance of unexpected objects), and commonsense (e.g., physically implausible object motions). Using this approach, the authors construct DualityVidQA, a paired dataset containing 144K realistic–counterfactual video pairs with automatically generated question–answer annotations. They further propose DNA-Train, a two-stage supervised fine-tuning (SFT) and reinforcement learning (RL) post-training procedure to teach MLLMs to detect anomalies (hallucinations). Experiments with Qwen2.5-VL-7B and LLaVA-Next-Video show substantial reductions in hallucinations across two hallucination-focused benchmarks, including one constructed by the authors, while maintaining or slightly improving performance on general-purpose video reasoning datasets.

**Strengths:**

- The paper tackles a crucial and timely problem of hallucination reduction in MLLMs by proposing a generative counterfactual augmentation strategy that is both intuitive and generalizable.
- The idea of constructing structured counterfactual perturbations at visual, semantic, and commonsense levels is original and nicely bridges data-centric and model-centric mitigation approaches.
- The proposed DNA-Train post-training strategy yields clear gains on multiple hallucination-oriented benchmarks, while avoiding catastrophic forgetting on general reasoning tasks.
- The overall framework is technically sound and well-motivated.

**Weaknesses:**

- The paper’s central hypothesis that hallucinations primarily arise from language priors within MLLMs is plausible but not convincingly validated in the context of this work. Although the results demonstrate reduced hallucinations, the paper lacks direct evidence showing that DualityForge specifically mitigates this root cause rather than merely benefiting from additional post-training. Including illustrative qualitative examples or control experiments would strengthen this argument.
- The category-wise analysis of counterfactuals (visual vs. semantic vs. commonsense) is missing. Since these categories likely engage different levels of reasoning (from perceptual recognition to causal inference), an ablation or per-category breakdown would provide valuable insight into what kinds of hallucinations the model actually learns to avoid.
- The use of the term “counterfactual” could be more rigorously justified, as some generated perturbations (e.g., brightness changes) may be better described as synthetic noise rather than true counterfactuals. Clarifying this conceptual distinction would improve precision.
- The paper overlooks recent hallucination evaluation benchmarks such as MHBench (Kong et al., 2025) and VidHalluc (Li et al., 2025), which are highly relevant for contextualizing the contribution and ensuring fair comparison.

Minor Issues:
- Verify the publication status of cited preprints (e.g., Zhang et al., 2025a appeared in ICLR 2025).

Missing References:
- Kong et al. MHBench: Demystifying Motion Hallucination in VideoLLMs. AAAI 2025.
- Li et al. VidHalluc: Evaluating Temporal Hallucinations in Multimodal Large Language Models for Video Understanding. CVPR 2025.

**Questions:**

1. Regarding your claim on mitigating language priors: how can we ensure that the observed gains stem from the counterfactual augmentation strategy, rather than simply from additional fine-tuning? Some baseline results comparing DNA-Train with standard SFT-only training would clarify this.
2. To strengthen empirical support, please evaluate on additional hallucination benchmarks (e.g., MHBench, VidHalluc) to better situate your improvements within the recent literature.
3. How were the general-purpose evaluation datasets selected? Did you follow a specific criterion (e.g., emphasizing temporal understanding or open-ended QA coverage)? Clarifying this would help assess the comprehensiveness of the evaluation setup.

---

### Official Review · Reviewer_BoRi · 2025-10-29

**Soundness:** 3
**Presentation:** 3
**Contribution:** 3
**Rating:** 4
**Confidence:** 4

**Summary:**

This paper proposes a large scale video (144k) QA dataset by visually augmenting the videos to tackle data imbalance problem. Subsequently, authors suggested a 2-staged (SFT+DAPO) DNA-Train with Duality Advantages Normalization that addresses falling into trivial solutions. Experiments show substantial gains in both hallucination and general video understanding benchmarks.

**Strengths:**

- The main idea of the DNA-Training strategy is mathematically sound and well-motivated for stable policy optimization and generalizable with other RL-training with binary rewards.
- Provides a solid problem statement that MLLMs relies on textual priors rather than visual inputs.
- This paper provides clear and logical ablation experiment settings and analysis that corroborates the claim of the authors, especially on dataset pairing, model scale, and DNA efficacy.

**Weaknesses:**

(Major)
- I am doubtful that such subtle video edits (blurring the frames in a short clip) can truly be counterfactual video contents to the original video.
- I understood the idea of Eq. 1 why such data samples are effective, adding short writings to explain the motivation would help the overall readability. Also does it indicate we are actually filtering data above/below the threshold? Or is it just a concept of the goal?
- There is a concern regarding circular annotation. AI generated synthetic data has a risk to encode the same underlying linguistic priors.
- Benchmark size is relatively small compared to existing literature. Also, I am curious how the performance would change if we scale or decrease the number of training sets. There might be further findings on this setting and comparing it to SFT only or RL only training configuration.
- How is the human verification process designed for Test split? Are DualityVidQA and DualityVidQA-RL also human verified?

(Minor)
- Inconsistent notation: DualityForge at line 77, DualityForce at Section 3.2 title and so on.
- The tested models are limited to the Qwen2.5-VL family. I am not necessarily asking the authors to perform this experiment, but would like to know if the proposed method generalizes well with other models, and other RL-training with binary rewards.


I will increase the score once current concerns are addressed.

**Questions:**

- Figure A1 is more helpful for understanding the data construction framework. Can authors consider moving it to the main script?
- Can’t we simply prompt generative models without visual input constraints for data generation pipeline? This may introduce bigger discrepancies of paired visual inputs.

---

### Official Review · Reviewer_LwFK · 2025-10-30

**Soundness:** 3
**Presentation:** 2
**Contribution:** 2
**Rating:** 4
**Confidence:** 4

**Summary:**

In video understanding tasks, models often over-rely on linguistic priors, leading to hallucinations. This issue arises from the imbalance between textual and visual data. To address this, the paper proposes a counterfactual data synthesis framework that generates a large number of counterfactual scenarios. Furthermore, to fully leverage the counterfactual data, the paper introduces a DNA-training framework, which enables more stable and effective policy optimization.

**Strengths:**

1. The proposed counterfactual data synthesis framework requires no human intervention, making it highly scalable.
2. The DNA training framework effectively leverages counterfactual data by performing contrastive learning on paired video samples, which helps the model reduce its reliance on linguistic priors and thereby mitigates hallucinations.

**Weaknesses:**

1. The counterfactual video generation process heavily relies on diffusion models, yet the temporal consistency control of these models remains unstable. The paper does not present any mechanism for controlling the quality of the generated videos (e.g., quantitative metrics such as FVD or CLIP Score).
2. The paper repeatedly claims improvements in “stability” and “balanced gradient updates,” but provides no mathematical derivations or convergence analysis—only empirical explanations—thus lacking rigorous theoretical support.
3. The core of “DualityForge” lies in using diffusion models to generate counterfactual videos and corresponding QA data. However, compared with existing frameworks combining video editing and synthetic QA generation (e.g., ShareGPT4Video, LLaVA-Video), its main innovations—“structured context embedding” and “contrastive QA”—are relatively limited. Moreover, “DNA-Train” merely adds l1 normalization to the advantage computation in the DAPO framework, which is essentially a minor optimization of training techniques.
4. The paper lacks comparisons with recent counterfactual generation or de-hallucination methods, such as Vision-R1[1], VLM-R1[3], and OpenVLThinker[2], which adopt RL-based approaches.
5. The ablation study does not examine the effect of l1 normalization on gradient variance or provide quantitative metrics of training stability.
6. Dataset validation is missing, making it difficult to ensure the visual plausibility of the edited videos. Furthermore, the automatically generated QA pairs may contain labeling errors, which might require human verification to assess the quality of the data generation pipeline.
7. Figure 2 is visually dense and lacks clear annotations, making it difficult to identify the key steps of DualityForge.

[1] Huang W, Jia B, Zhai Z, et al. Vision-r1: Incentivizing reasoning capability in multimodal large language models[J]. arXiv preprint arXiv:2503.06749, 2025.

[2] Deng Y, Bansal H, Yin F, et al. Openvlthinker: An early exploration to complex vision-language reasoning via iterative self-improvement[J]. arXiv preprint arXiv:2503.17352, 2025.

[3] Shen H, Liu P, Li J, et al. Vlm-r1: A stable and generalizable r1-style large vision-language model[J]. arXiv preprint arXiv:2504.07615, 2025.

**Questions:**

1. What are the actual costs of generating 140000 samples, including time, computing power, and editing success rate?

---

### Official Review · Reviewer_fzc9 · 2025-11-05

**Soundness:** 3
**Presentation:** 3
**Contribution:** 3
**Rating:** 4
**Confidence:** 4

**Summary:**

The paper tackles visual hallucination in Multimodal Large Language Models (MLLMs), where models rely on language priors instead of visual evidence when reasoning about counterfactual videos. The authors propose **DualityForge**, a controllable diffusion-based framework that edits real videos into counterfactual versions with structured contextual cues, enabling automatic generation of paired video–QA data. They further introduce **DualityVidQA**, a 144K-pair dataset for contrastive video QA, and **DNA-Train**, a two-stage (SFT + RL) training paradigm that applies ℓ₁-based *duality-normalized advantages* to balance learning between real and counterfactual samples. Experiments show significant reductions in hallucination without degrading general video understanding, achieving 76.8% pairwise accuracy on DualityVidQA-Test and competitive performance across multiple benchmarks.

**Strengths:**

**Originality.**
The paper introduces a novel perspective on mitigating *visual hallucinations* in MLLMs through counterfactual video synthesis — a rarely explored yet highly relevant direction. The integration of diffusion-based controllable video editing with structured contextual grounding is both creative and technically sound, expanding beyond conventional data balancing or textual augmentation methods.

**Quality.**
The work demonstrates a strong methodological design. The DualityForge pipeline is clearly defined, fully automated, and experimentally validated. The proposed DNA-Train training paradigm, particularly its ℓ₁-based *duality-normalized advantage* mechanism, is mathematically well-motivated and empirically effective, yielding consistent improvements across hallucination and general benchmarks.

**Clarity.**
The paper is well-written and logically structured. Figures and equations (especially Eq. (1)–(7)) effectively illustrate the conceptual and mathematical flow. The narrative maintains good coherence between the problem motivation, method design, and experimental validation.

**Significance.**
The contributions are practically important and theoretically meaningful. The construction of DualityVidQA provides a valuable benchmark for evaluating grounded reasoning in MLLMs, while DNA-Train offers a scalable training paradigm applicable to other multimodal tasks. Together, these innovations establish a solid step toward more visually grounded multimodal understanding.

**Weaknesses:**

**1. Data construction transparency and scale.**
The paper does not clearly quantify how many *unique* videos and edited counterparts underpin the 144K QA pairs, nor the average QA-per-video ratio and the real:synthetic balance. The editing configurations (e.g., duration, resolution, number of frames, edit categories per level) and the automatic QA generation settings are also under-specified. Without these essentials, DualityVidQA’s scalability and reproducibility are hard to assess, and downstream users cannot compare fairly to prior VLM/MLLM data regimes such as Flamingo-style video ingestion [1] or LLaVA-style instruction tuning [2].
A small table reporting per-split counts (unique videos, pairs, QA-per-video), edit parameters, and failure rates of edits/QA would materially improve clarity.

**2. Missing computational cost and throughput analysis.**
The framework couples diffusion-based video editing, automatic QA synthesis, SFT, and RL. Yet the paper omits:
(i) per-video editing time and success rate;
(ii) tokens/sec during QA synthesis;
(iii) SFT epoch time and RL sampling throughput (responses/prompt/sec); and
(iv) overall GPU-hours by stage.
Such numbers are standard for judging adoption cost and carbon/engineering budgets. Please report wall-clock numbers on the stated H200 setups and provide an efficiency comparison against common video-MLLM pipelines.

**3. Reward design remains ambiguous.**
The RL stage is described as “verifiable,” but it is unclear whether the reward checks *only* textual answer correctness or also enforces *visual grounding* (e.g., via caption or evidence alignment verifiers).
If rewards are purely answer-based, RL might reinforce language priors rather than visual grounding. A decomposition of the reward (correctness vs. format vs. grounding), plus ablations where each component is toggled, would strengthen the claim of reducing visual hallucination.
Prior RL-with-verifiable-rewards work (GRPO [4], DAPO [5]) discusses verifier design choices that could be cited and compared.


**4. Justification for SFT → RL ordering and algorithm choice.**
The paper fixes the training order to SFT then RL and adopts DAPO but does not argue *why* this order (vs. RL→SFT) or *why* DAPO over GRPO for this task.
Since training order affects convergence, exploration, and mode-collapse, a brief controlled study (SFT→RL vs. RL→SFT) and a head-to-head comparison (GRPO vs. DAPO) on the proposed tests would make the methodology more convincing.
The GRPO/RLVR literature emphasizes stability in verifiable tasks; it would be valuable to explain why DAPO is preferable for long-CoT video QA here.

**5. Limited analysis of *why* hallucinations drop.**
Beyond aggregate accuracy, there is no mechanistic evidence that visual grounding truly improves.
Recommended analyses include:
(a) cross-modal attention maps over training;
(b) modality-wise gradient norms;
(c) token-level attribution to frames vs. text; and
(d) attention or entropy diagnostics that have been used to study attention head roles and perception in LMs/VLMs [6].
These would connect DNA-Train’s ℓ₁ “duality-normalized advantage” to observable shifts in the model’s reliance on video signals.


**6. Benchmark dependence vs. structural limitation (language attention dominance).**
Even large counterfactual sets may not resolve the deeper, architecture-level issue where linguistic priors overshadow visual cues at inference—long documented from caption hallucination to instruction-tuned VLMs [3].
The paper should discuss to what extent DNA-Train *rebiases attention* rather than merely adapting to the benchmark distribution.
Consider adding stress tests (unseen counterfactual types, adversarial or low-quality edits) and reporting failure modes where plausibility still overrides visual evidence.


**7. Ablations and interpretability are too light for the claimed contributions.**
Key components (paired data, contrastive QA, SFT, RL, ℓ₁ normalization) are not sufficiently disentangled.
Provide ablations isolating each piece (and their interactions), plus qualitative case studies tracing how the *same* example evolves across training (e.g., before/after attention shift, evidence tokens used).
This would separate data-scale gains from genuine grounding effects.

**8. Generalization and robustness beyond video QA are unclear.**
Claims are centered on counterfactual video QA.
It remains unknown whether the approach extends to related tasks (temporal localization, retrieval), other modalities (image, audio-video, 3D), or noisy/OOD distribution settings.
A short transfer study or discussion of negative results would help bound the method’s scope.

**9. Human evaluation is missing.**
Because “hallucination” is partly perceptual, small-scale human studies (e.g., rating correctness, evidence grounding, and explanation helpfulness) would triangulate automatic metrics.
This is common in prior work on hallucination and relevance [3] and would make claims about *visual* grounding more credible.

---

### References

1. Alayrac, J.-B., et al. (2022). *Flamingo: A Visual Language Model for Few-Shot Learning.* NeurIPS.
2. Liu, H., Li, C., Wu, Q., Lee, Y. J. (2023). *Visual Instruction Tuning (LLaVA).* NeurIPS (Oral) / arXiv:2304.08485.
3. Rohrbach, A., et al. (2018). *Object Hallucination in Image Captioning.* EMNLP.
4. Mroueh, Y., et al. (2025). *Reinforcement Learning with Verifiable Rewards: GRPO’s Effective Loss Dynamics.* arXiv:2503.06639.
5. Yu, Q., et al. (2025). *DAPO: An Open-Source LLM Reinforcement Learning System at Scale.* arXiv:2503.14476.
6. *Unveiling Visual Perception in Language Models.* (2024). arXiv:2412.18108. (Attention head/perception analysis)

**Questions:**

> I encourage the authors to thoroughly address the weaknesses and questions raised in this review. If the authors can provide detailed explanations and in-depth clarifications during the rebuttal, and if the revised version demonstrates substantial progress in both clarity and improvement, I will be willing to reassess the manuscript and **adjust my overall rating** accordingly, based on the quality and depth of the revision.

---

**1. Data Construction Transparency and Scale**

* Could the authors specify the exact number of *unique* real and counterfactual videos used to generate the 144K QA pairs?
* What is the average number of QA pairs per video, and what is the ratio of real to synthetic samples?
* Please provide detailed editing parameters (e.g., frame rate, resolution, edit category taxonomy, diffusion prompt types) and QA generation latency statistics.
* Was any human verification applied during QA generation or post-edit filtering? If so, what proportion of samples were manually corrected or rejected?

**2. Computational Cost and Efficiency**

* How long does it take, on average, to synthesize a single counterfactual video and its corresponding QA pair?
* Could the authors provide a breakdown of GPU hours and throughput (responses/sec, tokens/sec) for each phase—diffusion editing, QA generation, SFT, and RL?
* How does the proposed pipeline compare in resource efficiency to other large-scale video instruction tuning frameworks such as LLaVA-Video or ShareGPT4Video?

**3. Reward Design and Verifiability**

* Does the reward function evaluate *only* textual correctness or also verify *visual grounding* (e.g., matching temporal regions, visual evidence, or captions)?
* How are conflicting signals handled when textual reasoning seems correct but contradicts visual content?
* Could the authors include ablations showing the effect of removing or modifying each reward component (correctness, format, grounding)?
* Are any external verifiers or discriminators used during RL for grounding validation?

**4. SFT → RL Order and Algorithm Choice**

* Why was the order fixed to SFT followed by RL, rather than experimenting with RL pretraining followed by SFT (which might stabilize reward learning)?
* What specific factors motivated the selection of DAPO instead of GRPO or PPO-based alternatives, especially regarding stability or entropy control?
* Did the authors attempt or observe any divergence when reversing the training sequence (RL→SFT)? If not, could this be tested?
* Could the authors share loss and reward trajectories to demonstrate that DAPO indeed avoids entropy collapse in this multimodal setting?

**5. Mechanistic Understanding of Hallucination Reduction**

* Beyond quantitative accuracy, can the authors visualize cross-modal attention maps before and after RL to demonstrate stronger video grounding?
* Have they measured per-token attribution or gradient norms across modalities to show that video signals receive higher relative weights?
* Is there empirical evidence linking the ℓ₁ “duality-normalized advantage” term to improved multimodal feature alignment?
* Would the authors consider incorporating diagnostic analyses (e.g., entropy, attention shift, visual feature coverage) to verify this mechanism?

**6. Benchmark Dependence and Architectural Limitations**

* Could the performance improvements stem from benchmark-specific adaptation rather than genuine robustness?
* How does the model behave under unseen or adversarial counterfactual edits (e.g., physically impossible events or low-quality videos)?
* Have the authors explored whether the language module still dominates decision-making attention during inference, as reported in prior perception studies?
* Could they consider measuring modality-wise attention weights during inference to confirm reduced “language attention dominance”?

**7. Ablation and Interpretability**

* Could the authors conduct systematic ablations isolating the effects of paired data, SFT, RL, and advantage normalization separately?
* Please include qualitative examples where the model’s reasoning or answer changes after each training phase.
* Would it be possible to visualize token-level or frame-level contributions to illustrate the internal grounding process?

**8. Generalization and Robustness**

* Have the authors tested DNA-Train on related tasks such as temporal localization, captioning, or retrieval?
* Does the method generalize to still images or multimodal datasets with different temporal dynamics?
* What happens when trained on a different base model (e.g., InternVideo, LLaVA-Next-Video)?
* Could the authors comment on out-of-distribution performance (e.g., unseen domains, distorted videos)?

**9. Human Evaluation**

* Have the authors considered conducting a small-scale human evaluation (e.g., via pairwise preference or grounding quality ratings)?
* How consistent are automatic hallucination metrics with human judgments of correctness and evidence grounding?
* Could the authors share examples where the model is “technically correct” but visually misleading?

---

### Note · Authors · 2025-11-13

I have read and agree with the venue's withdrawal policy on behalf of myself and my co-authors.